# Cardiac Rehabilitation for Older Women with Heart Failure

**DOI:** 10.3390/jpm12121980

**Published:** 2022-11-30

**Authors:** Domenico Scrutinio, Pietro Guida, Laura Adelaide Dalla Vecchia, Ugo Corrà, Andrea Passantino

**Affiliations:** 1Istituti Clinici Scientifici Maugeri SpA SB, IRCCS, Institute of Bari, 70100 Bari, Italy; 2Istituti Clinici Scientifici Maugeri SpA SB, IRCCS, Institute of Milan, 20138 Milan, Italy; 3Istituti Clinici Scientifici Maugeri SpA SB, IRCCS, Institute of Veruno, 28010 Veruno, Italy

**Keywords:** heart failure, cardiovascular rehabilitation, personalized treatment

## Abstract

Background: the role that sex plays in impacting cardiac rehabilitation (CR) outcomes remains an important gap in knowledge. Methods: we assessed sex differences in clinical and functional outcomes in 2345 older patients with heart failure (HF) admitted to inpatient CR. Three outcomes were considered: (1) the composite outcome of death during the index admission to CR or transfer to acute care; (2) three-year mortality; (3) change in six-minute walking distance (6MWD) from admission to discharge. Sex differences in outcomes were assessed using multivariable Cox or logistic regression models. Results: the hazard ratios of the composite outcome and of three-year mortality for females vs. males were 0.71 (95%CI:0.50–1.00; *p* = 0.049) and 0.68 (95%CI:0.59–0.79; *p* < 0.001), respectively. The standardized mean difference in 6MWD increase from admission to discharge between males and females was 0.10. The odds ratio of achieving an increase in 6MWD at discharge to values higher than the optimal sex-specific thresholds for predicting mortality for females vs. males was 2.21 (95%CI:1.53–3.20; *p* < 0.001). Conclusion: our findings suggest that older females with HF undergoing CR have better prognosis and garner similar improvement in 6MWD compared with their male counterparts. Nonetheless, females were more likely to achieve levels of functional capacity predictive of improved survival.

## 1. Introduction

Heart failure (HF) is an increasingly prevalent clinical syndrome burdened by high mortality and morbidity rates [1,2]. Functional disability and loss of independence are hallmark features of HF. The Global Burden of Disease study showed that HF is responsible for a significant proportion of all years lived with disability among people with chronic diseases [3]. Sex differences are emerging as an increasingly important issue in cardiovascular medicine. As noted by Lam et al. [4], some of the most profound sex differences are found in HF. Sex differences in demography, etiology, pathophysiology, cardiac structure and function, functional capacity and quality of life have been consistently documented in HF patients [4].

Mounting evidence indicates that cardiac rehabilitation (CR) can significantly improve symptoms, functional status and quality of life, can reduce hospital admissions and mortality, and is helpful to promote treatment optimization and monitor symptoms in patients with HF [5,6,7,8,9,10,11]. Although CR is a class 1A recommendation for patients with HF [12,13], it is underutilized, especially in women and older patients [14]. More than 80% of HF patients are 65 years of age or over [15]. Compared with younger patients, older patients more often are females and have preserved left ventricular ejection fraction (LVEF), higher burden of comorbidities, poorer functional capacity and poorer prognosis, thus representing a particularly challenging population in CR [16]. Better understanding of sex differences in this patient population could inform initiatives to reduce potential sex disparities and to improve outcomes in the CR setting. While barriers to CR and sex disparities in the rates of referral to and enrolment in CR have been extensively studied [17], the role that sex plays in impacting CR outcomes remains an important gap in knowledge [18]. In this study, we examined sex differences in functional and clinical outcomes in a large cohort of older patients with HF admitted to inpatient CR.

## 2. Material and Methods

This was a multicenter observational retrospective study including all discharges with a primary diagnosis of HF (International Classification of Diseases, Ninth Revision codes: 402.01, 402.11, 402.91, 404.01, 404.03, 404.11, 404.13, 404.91, 404.93, and 428.xx) from six inpatient CR units of a nationwide Research Institute in the field of Rehabilitation Medicine in Italy between January 2013 and December 2016 (6). All participating centers are part of a single department of CR and share a common formal rehabilitation program. According to the national regulatory rules governing admissions to inpatient CR for HF in Italy [19], patients were admitted just after a hospitalization for HF or because of declining functional capacity and/or deteriorating clinical status. During the study period, 3301 patients experienced 5312 admissions. In the case of multiple admissions, the first admission was selected for inclusion in the study. Patients were eligible for inclusion in the study if they were ≥65 years of age. Of the 3301 patients, 2345 met the eligibility criterion and were included in the study.

As previously reported in a companion article (6), our formal multidisciplinary program is led by cardiologists and is designed to promote stable clinical conditions, improve physical function through a supervised exercise training plan tailored to the individual level of functional ability at presentation, provide specialized medical assistance, and optimize medical treatment. The exercise program consists of a supervised training program including active/passive mobilization; assisted ambulation; respiratory, musculoskeletal flexibility, movement coordination and/or callisthenic exercises and training on a (unloaded) bedside/upright cycle ergometer. The types of exercises and exercise intensity are gradually progressed throughout the rehabilitation period, according to the individual functional and clinical conditions. Experienced physiotherapists perform a standardized 6-min walking test (6MWT) at admission to and discharge from CR [20].

### 2.1. Data Collection

Data collection procedure was previously described (6). The data were extracted from the electronic hospital information system shared between the participating centers and entered into a REDCap database. Baseline measurements were obtained at the time of admission to inpatient CR. All patients provided written consent to the use of their data in an anonymous form for scientific purposes. Any identifying information was removed from the database and replaced with an identification number. Survival status was ascertained by linkage to the national health information system. The patients were followed-up until death or 30 November 2019.

### 2.2. Outcomes

Three outcomes were considered. First, we analyzed sex differences in the incidence of the composite outcome of death during the index admission to inpatient CR or transfer from inpatient CR to acute care. Deaths that occurred within 24 h from discharge were counted as in-hospital deaths. Then, we analyzed sex-differences in 3-year all-cause mortality after discharge from CR. Finally, we analyzed sex differences in functional outcome, based on distance walked on 6MWT (6MWD). To evaluate a clinically relevant threshold of improvement in 6MWD from admission to discharge, two measures were used. First, we defined an outcome of 6MWD based on absolute change from admission to discharge; a priori, we chose the highest quartile of change in 6MWD in the overall cohort as the binary threshold [21]. Second, we compared the rate of achievement of 6MWD values at discharge higher than the optimal threshold of 6MWD for predicting mortality. If a patient was not able to walk, the 6MWD was set to 0 m [22].

### 2.3. Statistical Analysis

Data are reported as mean and standard deviation (SD) or median with 25th and 75th percentiles for continuous variables and as number and percentage for categorical variables. We used the Student’s *t*-test or the Mann–Whitney test to compare continuous variables and the χ^2^ test to compare categorical variables. Cumulative survival was estimated using the Kaplan–Meier method and a log-rank test was used to compare groups.

#### 2.3.1. Composite Outcome

Crude and adjusted hazard ratios (HR) of the composite outcome for females compared with males were estimated by using unadjusted and adjusted Cox proportional hazards models. The regression models included the following covariates: age, obesity, hypertension, diabetes mellitus, chronic obstructive pulmonary disease (COPD), atrial fibrillation, transfer to CR from acute care hospitals after a hospitalization for HF, New York Heart Association (NYHA) class III/IV, systolic blood pressure, LVEF, estimated glomerular filtration rate (eGFR) and hemoglobin and sodium levels. Missing data for systolic blood pressure (3.6%), eGFR (0.4%), hemoglobin (0.7%) and sodium levels (0.5%) were replaced by the median of observed values [23]. Cox regression analyses were repeated in the subset of patients with available data for N-terminal pro-B-type natriuretic peptide (NT-proBNP) (436 females and 937 males).

#### 2.3.2. Three-Year Mortality

Crude and adjusted hazard ratios (HR) of mortality for females compared with males were estimated by using unadjusted and adjusted Cox proportional hazards models. The regression models included age, obesity, hypertension, diabetes mellitus, COPD, atrial fibrillation, transfer to CR from acute care hospitals after a hospitalization for HF, NYHA class III/IV, systolic blood pressure, LVEF, eGFR, hemoglobin and sodium levels, and transfer from CR to acute care. These variables were selected because they were identified in previous studies as being strong predictors of mortality [24,25]. Cox regression analyses were repeated in the subset of patients with available data for NT-proBNP.

#### 2.3.3. Functional Outcome

We computed the standardized mean difference in 6MWD change from admission to discharge between females and males. Multivariable logistic regression models were used to assess the impact of female sex on the two measures of functional outcome. Since females are known to have lower functional capacity compared with males, sex-specific thresholds of 6MWD at discharge for predicting mortality were identified. The optimal threshold was considered as the one that maximized the chi-square statistic. The regression models included age, obesity, hypertension, diabetes mellitus, COPD, atrial fibrillation, transfer to CR from acute care hospitals after a hospitalization for HF, NYHA class, systolic blood pressure, LVEF, hemoglobin, eGFR and 6MWD at admission [26,27]. The presence of an interaction between sex and age, NYHA III/IV class, LVEF and 6MWD at admission was also tested using the likelihood ratio test. The analyses were conducted using STATA software, version 14 (Stata-Corp LP, College Station, Tex).

## 3. Results

Figure 1 displays the flow-chart of the study.

F denotes females, M males, CR cardiac rehabilitation, IRF inpatient rehabilitation facility, 6MWT 6-min walk test.

Table 1 displays patient baseline characteristics stratified by sex.

There were some noteworthy differences between males and females. Females were significantly older and more often had hypertension, atrial fibrillation and preserved EF. Females also had more prevalent and severe chronic kidney disease, lower hemoglobin levels, poorer functional capacity and more often presented with total/severe dependence in performing activities of daily living than males.

### 3.1. Composite Outcome

A total of 77 patients (24 females [2.7%], 53 males [3.6%]) died during the index admission to CR and 106 (28 females [3.2%], 78 males [5.3%]) were transferred to acute care from CR (Figure 1).

The composite outcome occurred in 5.9% of females and 8.9% of males (*p* = 0.008). Crude HR of the composite outcome for females compared with males was 0.61 (95% confidence intervals [CI] 0.44–0.84; *p* = 0.003). Following full adjustment for covariates, the HR was 0.71 (95% CI 0.50–1.00; *p* = 0.049). After further adjustment for NT-proBNP, the HR was 0.58 (95% CI 0.36–0.93; *p* = 0.023) (Table 2).

### 3.2. Three-Year All-Cause Mortality

A total of 2268 patients were discharged alive (Figure 1). Of these patients, 43 (1.9%) were lost to follow-up leaving 2225 patients (844 females and 1381 males) available for survival analysis (Figure 1). A total of 4766 person-years of follow-up were examined during which 993 deaths (20.8 deaths/100 person-years) occurred. Crude HR of mortality for females compared with males was 0.79 (95% CI 0.70–0.91; *p* = 0.001). Following full adjustment for covariates, the HR was 0.68 (95% CI 0.59–0.79; *p* < 0.001). After further adjustment for NT-proBNP, the HR was 0.65 (95% CI 0.53–0.79; *p* < 0.001) (Table 2).

### 3.3. Functional Outcome

A total of 2102 patients were discharged home (Figure 1). Of these patients, 1607 (594 females, 1013 males) had paired data for 6MWD at admission and discharge (Figure 1). Mean 6MWD at admission was significantly lower in females than in males (137 ± 132 vs. 224 ± 158; *p* < 0.001). At discharge, 6MWD increased by 53 ± 71 m in females and by 61 ± 86 m in males (*p* = 0.065). The standardized mean difference in 6MWD increase between males and females was 0.10 (95% CI 0.02–0.18).

In the entire cohort, the 75th percentile of the change in 6MWD from admission to discharge was 85 m. The crude OR of achieving an increase in 6MWD >85 m for females compared with males was 0.97 (95% CI 0.77–1.23; *p* = 0.800). Following adjustment for covariates, the OR was 0.80 (95% CI 0.62–1.05; *p* = 0.117) (Table 2). No significant interaction between sex and selected covariates was found.

The sex-specific optimal thresholds of discharge 6MWD for predicting mortality were 225 m for females and 287 m for males. At admission, 418 (70.4%) females and 591 (58.3%) males had 6MWD values below the optimal threshold. Among these patients, the adjusted odds ratio of achieving an increase in 6MWD at discharge to values higher than the optimal threshold for females compared with males was 2.21 (95% CI 1.53–3.20; *p* < 0.001). Figure 2 displays survival curves for females and males who did or did not achieve an increase in 6MWD at discharge to values higher than the optimal threshold.

Survival curves were constructed using the Kaplan–Meier Survival Analysis calculator [Internet]. Statistics Kingdom 2017. Available from: https://www.statskingdom.com/kaplan-meier.html (accessed on 19 July 2022).

At multivariable Cox regression analysis, failure to achieve discharge 6MWD values exceeding the optimal threshold was associated with markedly increased mortality risk both in females (HR 2.64 [95% CI 1.88–3.70]; *p* < 0.001) and in males (HR 2.07 [95% CI 1.66–2.59]; *p* < 0.001).

Of the 2102 patients discharged home, 364 had missing data for 6MWT at admission and 131 had missing data for 6MWT at discharge. For these patients, no information on why the 6MWT was not performed was available. Compared with both the included patients and those with missing data for 6MWT at discharge, the patients with missing data for 6MWT at admission had significantly poorer survival (Appendix A).

## 4. Discussion

Heart failure patients participating in CR research tend to be relatively young, predominantly males and with a low burden of comorbidities, thus limiting the transferability of research findings to the general HF population. In the ExTra-MATCH II Meta-Analysis, for example, the mean age was 61 years and only 28% of the participants were females [28]. Norris et al. recognized underrepresentation of women in cardiovascular research as a “barrier to generating knowledge and developing clinical practice guidelines” [29]. Three major findings emerged from this study. First, females had a lower risk of early adverse clinical outcome and long-term mortality compared with males. Second, no significant difference in the extent of functional improvement following CR between females and males was observed. Third, females were more likely to achieve an increase in 6MWD at discharge to values higher than the optimal threshold for predicting mortality, compared with males.

Females had better prognosis than males. After full adjustment for well-established prognostic factors, females had a 29% lower risk of early adverse outcome and a 32% lower risk of death within 3 years, compared with their male counterparts. A female prognostic advantage in HF has been observed in previous studies. In a population-based cohort study of people with incident HF, age-adjusted mortality after HF diagnosis was 33% higher in males than in females [30]. In the MAGGIC individual patient meta-analysis, survival was better for females compared with men, irrespective of LVEF [31]. In the CHARM Program, which included a broad spectrum of patients with chronic HF, females had lower risks of most fatal and nonfatal outcomes that were not explained by LVEF, origin of the HF, or adherence to therapy [32]. In a clinical trial of advanced HF, females had better survival than their male counterparts [33]. Finally, in a registry of HF patients, females showed better survival than males across the EF spectrum, after extensive adjustments [34]. The present study adds to the existing knowledge by showing that the survival advantage for females extends to the rehabilitation setting. The biological mechanisms underlying the gender gap in survival in HF remain elusive. However, some observations may help the interpretation of the female survival advantage observed in the present and previous studies. First, as observed by Austad [35], females are less likely to succumb to most of the major causes of death. Second, studies suggest that the prognostic benefit of CR is greater in females than in males [36,37]. In the HF-ACTION trial [36], a significant 26% reduction in the primary endpoint of all-cause mortality or all-cause hospital stay in females and no reduction in males were observed. Third, recovery of LVEF from HF with reduced LVEF, which is strongly linked to improved survival [38], is more common in females than in males [39]. Finally, in a population-based study, females were significantly less likely to progress to advanced HF than males, after extensive adjustment for covariates [40].

The patients’ baseline characteristics are indicative of a cohort of older, vulnerable patients with severely impaired functional capacity. In such patients, a 6MWT may represent maximal effort [41] and its responsiveness is greatest [42]. Despite a poorer functional status at admission, females garnered similar improvement in functional capacity compared with males. Indeed, no significant difference in the extent of 6MWD increase from admission to discharge between females and males was observed. The standardized mean difference in 6MWD increase between males and females was 0.10, indicating negligible difference. The thresholds of 225 m for 6MWD at discharge in females and of 287 m in males demonstrated the strongest association with mortality. The achievement of a 6MWD at discharge higher than these thresholds predicted markedly improved survival in both sexes. Females were >2 times more likely to achieve such an improvement in distance walked than males. These findings underscore the role of 6MWD as a key outcome measure for older patients with HF admitted to CR, suggest that the prognostic value of 6MWT should be assessed separately in males and females, and may have implications to explain the observed survival advantage for females.

We focused on sex-related differences in clinical and functional outcomes. While sex refers to biological characteristics, gender refers to “psychological, social, and cultural factors that shape attitudes, behaviors, and knowledge” [43]. In contrast to sex, gender is not a binary term; as observed by Mauvais-Jarvis et al. [44], indeed, “traits of masculinity and femininity most often coexist, are expressed to different degrees and are dynamic”. As such, gender can influence health and disease outcomes differently than biological sex. Thus, integrating sex- and gender-based analysis can lead to improved assessment and knowledge of differences in disease outcomes [43].

### Limitations

Limitations were previously acknowledged [6,11,45]. First, the study was retrospective in nature and a referral bias cannot be excluded. Second, the patients’ baseline characteristics are indicative of an elderly cohort with severely impaired functional capacity admitted to inpatient CR. This limits the generalizability of the results. Third, although we adjusted for well-established prognostic factors, other unmeasured factors might have influenced the association between sex and clinical outcomes. Fourth, 364 patients had missing data for 6MWT at admission and 131 at discharge. For these patients, no reason for not performing the 6MWT was recorded. Compared to the patients with available paired data for 6MWT, those with missing data for 6MWT at admission had markedly poorer 3-year survival, suggesting greater HF severity with possible inability to perform a 6MWT. Conversely, there was no difference in survival between patients with missing data for 6MWT at discharge and those with available paired data. Fifth, the retrospective nature of the study did not allow us provide granular detail on the CR delivered. Thus, we could not address potential differences in the dose-effect relationship of CR and functional outcome between females and males. Finally, sex differences in CR programming and outcomes can vary across countries as significant differences exist in *“CR availability, how CR is incorporated into the health system, core components of CR programming, and CR delivery models*” [18].

## 5. Conclusions

Our findings suggest that older females with HF undergoing inpatients CR have better prognosis compared with their male counterparts. Despite a poorer functional status at admission, females garnered similar absolute improvement in functional capacity following CR compared with males. Nonetheless, among the patients with more severe functional impairment at presentation, females were significantly more likely to achieve levels of functional capacity predictive of improved survival than males. Our findings support current efforts to promote referral and participation of females with HF in CR.

## Figures and Tables

**Figure 1 jpm-12-01980-f001:**
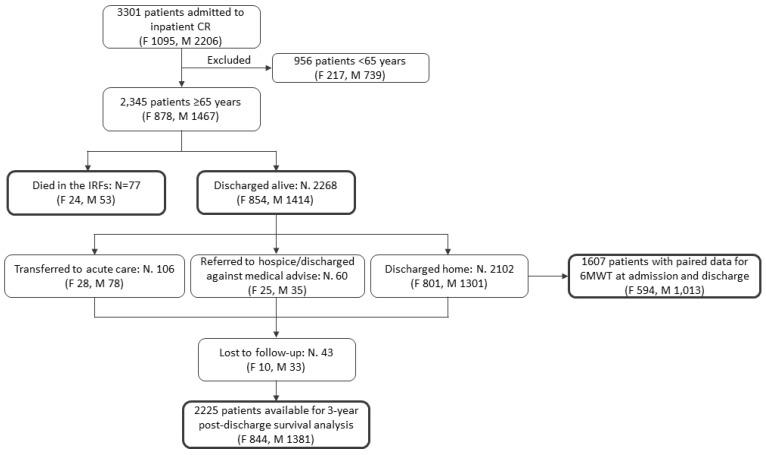
Flow-chart of the study.

**Figure 2 jpm-12-01980-f002:**
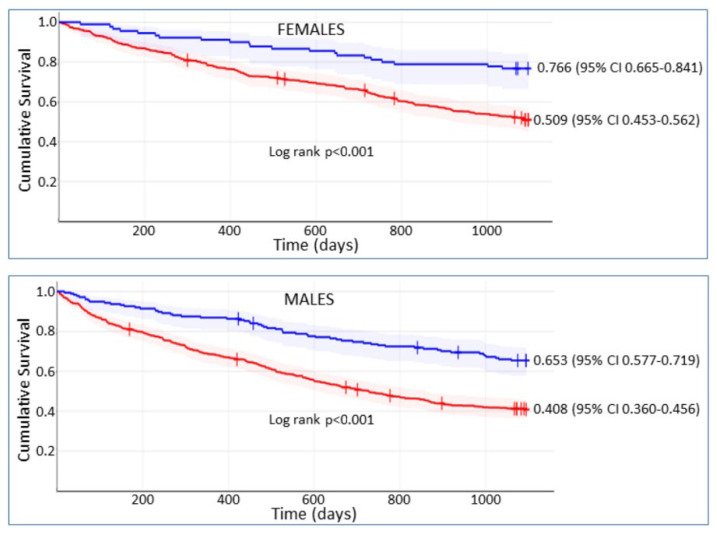
Survival curves for females and males who did (blue lines) or did not achieve (red lines) an increase in 6-min walk distance at discharge to values higher than the optimal threshold for predicting mortality.

**Table 1 jpm-12-01980-t001:** Baseline characteristics.

	Females (*n* = 878)	Males (*n* = 1467)	*p* Value
Demographics			
Age (years), mean (SD)	79 (7)	76 (7)	<0.001
Age > 75 years, *n* (%)	624 (71.1)	786 (53.6)	<0.001
Comorbidities			
Obesity (body mass index ≥ 30), *n* (%)	188 (21.4)	311 (21.2)	0.903
Hypertension, *n* (%)	727 (82.8)	942 (64.2)	<0.001
Diabetes mellitus, *n* (%)	266 (30.4)	519 (35.5)	0.012
Chronic obstructive pulmonary disease, *n* (%)	197 (22.4)	447 (30.5)	<0.001
Chronic kidney disease, *n* (%)	646 (74.3)	984 (67.1)	<0.001
Stage 3a (eGFR 45–59 mL/min/1.73 m^2^)	188 (21.6)	373 (25.4)	<0.001
Stage 3b (eGFR 30–44 mL/min/1.73 m^2^)	253 (29.1)	385 (26.3)	
Stage 4 (eGFR 15–29 mL/min/1.73 m^2^)	180 (20.7)	207 (14.1)	
Stage 5 (eGFR < 15 mL/min/1.73 m^2^)	25 (2.9)	19 (1.3)	
Anemia (hemoglobin < 13 g/dL in men and <12 g/dL in women), *n* (%)	485 (56.0)	855 (58.5)	0.243
Atrial fibrillation, *n* (%)	463 (52.9)	653 (44.5)	<0.001
Clinical findings			
Transferred from acute care hospitals after a hospitalization for HF, *n* (%)	418 (47.6)	672 (45.8)	0.398
NYHA III/IV class, *n* (%)	474 (54)	822 (56.0)	0.334
ICD/CRT-D, *n* (%)	79 (9.0)	354 (24.1)	<0.001
ICD/CRT-D in patients with LVEF ≤ 0.40, *n* (%)	64 (20.7)	317 (31.5)	<0.001
Systolic blood pressure (mm Hg), mean (SD)	117 (17)	113 (17)	<0.001
Systolic blood pressure < 100 mm Hg, *n* (%)	80 (9.6)	227 (16.2)	<0.001
Diastolic blood pressure (mm Hg), mean (SD)	69 (9)	68 (9)	0.009
Left ventricular ejection fraction			
Mean (SD)	47 (14)	36 (13)	<0.001
≥0.50, *n* (%)	443 (50.5)	287 (19.6)	<0.001
0.41–0.49, *n* (%)	126 (14.4)	173 (11.8)	
≤0.40, *n* (%)	309 (35.2)	1007 (68.6)	
Laboratory findings			
Hemoglobin (g/dL), mean (SD)	11.6 (1.8)	12.4 (2.0)	<0.001
Creatinine (mg/dL), (mean (SD)	1.35 (0.63)	1.58 (0.74)	<0.001
eGFR (mL/min/1.73 m^2^), mean (SD)	47 (22)	52 (23)	<0.001
Sodium (mEq/L), mean (%)	139.4 (3.7)	139.0 (3.8)	0.015
Sodium < 136 mEq/L, *n* (%)	116 (13.4)	235 (16.0)	0.080
NT-proBNP (pg/mL), median (IQR)	2985 (1052–5988)	2995 (1162–6104)	0.257
Functional status			
Barthel index at admission, mean (SD)	69 (26)	74 (26)	<0.001
Barthel index at admission ≤ 60 (total/severe dependence), *n* (%)	231 (35.5)	276 (25.8)	<0.001
Six-min walking distance at admission (meters), mean (SD)	137 (132)	224 (158)	<0.001
Length of stay in the IRFs (days), mean (SD)	22 (16)	21 (11)	0.001
Evidence-based treatments (patients with LVEF ≤ 0.40 discharged home)			
Number of patients	284	899	
RAAS-Is, *n* (%)	220 (77.5)	708 (78.8)	0.645
Beta-blockers	248 (87.3)	822 (91.4)	0.039
RAAS-Is plus beta-blockers	197 (69.4)	656 (73.0)	0.237

Abbreviations. CRT—cardiac resynchronization therapy, eGFR—estimated glomerular filtration rate, ICD—implantable cardioverter defibrillator, LVEF—left ventricular ejection fraction, NYHA—New York Heart Association, *n*—number of patients, RAAS-Is—renin angiotensin aldosterone system inhibitors, SD—standard deviation.

**Table 2 jpm-12-01980-t002:** Results of multivariable analyses.

Outcomes	HR (95% CI) Females vs. Males	*p* Value
Composite outcome	0.71 (0.50–1.00)	0.049
Three-year mortality	0.68 (0.59–0.79)	<0.001
Functional outcome	OR (95% CI) Females vs. males	
Increase in 6MWD to the highest quartile of change from admission to discharge in the overall cohort	0.80 (0.62–1.05)	0.117
Increase in 6MWD at discharge to values higher than the optimal threshold to predict mortality	2.21 (1.53–3.20)	0.001
Patients with available data for NT-proBNP		
	HR (95% CI) Females vs. males	
Composite outcome	0.58 (0.36–0.93)	0.023
Three-year mortality	0.65 (0.53–0.79)	<0.001

Abbreviations: HR—denotes hazard ratio, OR—odds ratio, 6MWD—six-minute walking test, NT-proBNP—N-terminal pro-B-type natriuretic peptide.

## Data Availability

Not applicable.

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
