# Peer review of "Cardiac Rehabilitation for Older Women with Heart Failure"

_jpm, 2022, doi:10.3390/jpm12121980_

Round 1
Reviewer 1 Report
This is an interesting multicenter observational retrospective study including patients with a primary diagnosis of Heart Failure. The topic is relevant and is related to gender differences in functional and clinical outcomes in a significant number of elderly patients (2,345) with heart failure admitted to 13 rehabilitation centers distributed throughout Italy. The data suggest that older females with heart failure undergoing inpatients Cardiac Rehabilitation have better prognosis compared with male patients underlining the usefulness of promoting residential rehabilitation programs in women with heart failure.
In order to publish the paper in Journal of Personalized Medicine we suggest editing and spelling in English using native speakers.
Author Response
REVIEWER 1
This is an interesting multicenter observational retrospective study including patients with a primary diagnosis of Heart Failure. The topic is relevant and is related to gender differences in functional and clinical outcomes in a significant number of elderly patients (2,345) with heart failure admitted to 13 rehabilitation centers distributed throughout Italy. The data suggest that older females with heart failure undergoing inpatients Cardiac Rehabilitation have better prognosis compared with male patients underlining the usefulness of promoting residential rehabilitation programs in women with heart failure.
In order to publish the paper in Journal of Personalized Medicine we suggest editing and spelling in English using native speakers.
Response.
We thank the reviewer for his/her review.
The language has been revised by a professional English language expert.
Reviewer 2 Report
The paper deals with a very interesting topic: cardiac rehabilitation for older women with heart failure. As noted by the authors "the role that sex plays in impacting cardiac rehabilitation outcomes remains an important gap in knowledge". Therefore, better understanding of sex differences in this patient population could inform initiatives to reduce potential sex disparities and to improve outcomes in the cardiac rehabilitation setting.
I believe that the article is well written, the authors chose the appropriate statistical methods and presented the obtained results in an understandable way.
Thank you for the opportunity to review this well constructed study. However, I have some suggestions for improving the manuscript:
1. The abstract should not be in the form of a short story - please correct the layout of the abstract and separate the following parts: introduction, aim of the study, material and methods, results, conclusions.
2. In their manuscript, the authors examined sex differences in a cohort of older HF patients - can they relate to younger people (<65 years of age), how do they correlate with them?
3. In the Material and Methods subchapter, please add information about the inclusion and exclusion criteria for this study - (it is not clear whether age was the only criterion) and what about the clinical condition of the patients? comorbidities? EF-value?
4. What were the values ​​of the ejection fraction of the examined patients, before and after rehabilitation?- please add.
5. In the presented manuscript, it is not clear what the exercise/rehabilitation program was - please add a separate section describing the exercise program - what was their duration? what was the duration of training on the cycle ergometer?
6. Were the exercises performed daily from Monday to Sunday? or every other day? or was it exercise one day and bike the next?.
7. How long (days) did the rehabilitation last? were they weekly cycles?
8. How was the intensity of exercise (and the bike) increased/and on what basis?
9. Were women or men more intense in the study?
10. In the manuscript, please add the approval number of the ethics committee;
11. Under Table 2, please add descriptions of the abbreviations used (HR, OR, CI, NT-proBNP, 6MWD)
12. In subsection 2.3.1, please expand the abbreviations - NYHA, LVEF, NT-proBNP;
13. In subsection 2.3.2, please also expand the abbreviation COPD;
14. At the beginning of the Discussion, the authors write: " Heart failure patients participating in CR research tend to be relatively young, predominantly males, and with a low burden of comorbidities, ..." - what did the author mean by "relatively young" - please define yourself.
Author Response
REVIEWER 2
The paper deals with a very interesting topic: cardiac rehabilitation for older women with heart failure. As noted by the authors "the role that sex plays in impacting cardiac rehabilitation outcomes remains an important gap in knowledge". Therefore, better understanding of sex differences in this patient population could inform initiatives to reduce potential sex disparities and to improve outcomes in the cardiac rehabilitation setting.
I believe that the article is well written, the authors chose the appropriate statistical methods and presented the obtained results in an understandable way.
Thank you for the opportunity to review this well constructed study. However, I have some suggestions for improving the manuscript:
COMMENT 1.
The abstract should not be in the form of a short story - please correct the layout of the abstract and separate the following parts: introduction, aim of the study, material and methods, results, conclusions.
Response 1.
Thank you for this comment. According to the instructions for author of the Journal, the abstract has been separated into the following parts: background, methods, results and conclusion.
COMMENT 2.
In their manuscript, the authors examined sex differences in a cohort of older HF patients - can they relate to younger people (<65 years of age), how do they correlate with them?
Response 2.
Thank you for this comment. Indeed, comparing older with younger patients undergoing cardiac rehabilitation could be of interest. However, this was not the aim of our study.
In the introduction section, we stated, “Compared with younger patients, older patients more often are females and have preserved left ventricular ejection fraction (LVEF), higher burden of comorbidities, poorer functional capacity and poorer prognosis, thus representing a particularly challenging population in CR [16]”.
COMMENT 3.
In the Material and Methods subchapter, please add information about the inclusion and exclusion criteria for this study - (it is not clear whether age was the only criterion) and what about the clinical condition of the patients? comorbidities? EF-value?
Response 3.
As shown in the flow-chart of the study (Figure 1), 956 patients aged less than 65 years were excluded. Age was the only criterion for exclusion. Comorbidities and EF values were reported in table 1.
COMMENT 4.
What were the values ​​of the ejection fraction of the examined patients, before and after rehabilitation?- please add.
Response 4.
The values of ejection fraction at admission to cardiac rehabilitation were reported in table 1.
Repeated measurement of LVEF at discharge was not performed.
In clinical trials of exercise training in heart failure, improvements in LVEF were trivial and did not reach statistical significance. On the other hand, it is well established that exercise training is not associated with adverse effects on left ventricular remodeling and function (Bozkurt B, et al. Cardiac Rehabilitation for Patients With Heart Failure: JACC Expert Panel. J Am Coll Cardiol. 2021;77:1454-1469)
COMMENT 5. In the presented manuscript, it is not clear what the exercise/rehabilitation program was - please add a separate section describing the exercise program - what was their duration? what was the duration of training on the cycle ergometer? COMMENT 6. Were the exercises performed daily from Monday to Sunday? or every other day? or was it exercise one day and bike the next?. COMMENT 7. How long (days) did the rehabilitation last? were they weekly cycles? COMMENT 8. How was the intensity of exercise (and the bike) increased/and on what basis? COMMENT 9. Were women or men more intense in the study?
Responses 5 to 9
We thank the Reviewer for these comments.
Some points need to be considered. First, the setting was inpatient rehabilitation. Second, the inpatient setting did allow maximize the adherence to the rehabilitation program. Third, according to the regulatory rules governing admission to inpatient cardiac rehabilitation, the total time of the rehabilitation activities was at least 500 minutes per week, from monday to saturday. Conformity with this standard is subject to periodic external audit by independent auditors of the Regional Health Agencies. Because of the retrospective design of the study, we could not provide details about progression to higher functional levels during the course of the intervention. The mean length of stay was 22±16 days for females and 21±11 days for men (table 1).
In the limitations section we reported, “Fifth, since this was a retrospective study, we could not provide granular detail on the CR delivered.Thus, we could not address potential differences in the dose-effect relationship of CR and functional outcome between females and males”.
COMMENT 10.
In the manuscript, please add the approval number of the ethics committee;
Response 10.
The approval number of the ethics committee has been reported in the “Institutional Review Board Statement” section.
COMMENT 11.
Under Table 2, please add descriptions of the abbreviations used (HR, OR, CI, NT-proBNP, 6MWD)
Response 11.
Done
COMMENT 12.
In subsection 2.3.1, please expand the abbreviations - NYHA, LVEF, NT-proBNP;
Response 12.
Done. Note, LVEF had been expanded in the Introduction section, line 41.
COMMENT13.
In subsection 2.3.2, please also expand the abbreviation COPD;
Response 13
The acronym COPD had been expanded in the preceding subsection 2.3.1
COMMENT 14. At the beginning of the Discussion, the authors write: " Heart failure patients participating in CR research tend to be relatively young, predominantly males, and with a low burden of comorbidities, ..." - what did the author mean by "relatively young" - please define yourself.
Response 14
We have added the following sentence in the discussion section:
In the ExTra-MATCH II Meta-Analysis, for example, mean age was 61 years and only 28% of the participants were females [29].
The following reference was added:
[29] Taylor RS, Walker S, Smart NA, Piepoli MF, Warren FC, Ciani O, Whellan D, O'Connor C, Keteyian SJ, Coats A, et al. Impact of Exercise Rehabilitation on Exercise Capacity and Quality-of-Life in Heart Failure: Individual Participant Meta-Analysis. J Am Coll Cardiol. 2019;73:1430-1443.